# Image Enhancement Method for Photoacoustic Imaging of Deep Brain Tissue

**Yonghua Xie [1], Dan Wu [1,\*], Xinsheng Wang [1], Yanting Wen [1], Jing Zhang [1], Ying Yang [1], Yi Chen [1], Yun Wu [1], Zihui Chi [1] and Huabei Jiang [2,\*]**

1    School of Optoelectronic, Chongqing University of Posts and Telecommunications, Chongqing 400065, China; s210401033@stu.cqupt.edu.cn (Y.X.); s220401035@stu.cqupt.edu.cn (X.W.); d200201024@stu.cqupt.edu.cn (Y.W.); d220201040@stu.cqupt.edu.cn (J.Z.); d210201032@stu.cqupt.edu.cn (Y.Y.); d200201003@stu.cqupt.edu.cn (Y.C.); d230201039@stu.cqupt.edu.cn (Y.W.); chizh@cqupt.edu.cn (Z.C.)

2    Department of Medical Engineering, University of South Florida, Tampa, FL 33620, USA

\*    Correspondence: wudan@cqupt.edu.cn (D.W.); hjiang1@usf.edu (H.J.)

**Abstract:** Photoacoustic imaging (PAI) is an emerging biomedical imaging modality, offering numerous advantages, including high resolution and high contrast. In its application to brain imaging, however, the photoacoustic (PA) signals from brain tissue weaken considerably due to the distortion effects of the skull. This attenuation reduces the resolution and contrast significantly. To address this issue, here we describe a Log-MSR algorithm that combines the logarithmic depth logarithmic enhancement (Log) algorithm and the multi-scale Retinex (MSR) algorithm. In this method, the Log algorithm performs local weighted compensation based on signal attenuation for different depths, while the MSR algorithm improves the contrast of the image. The proposed Log-MSR algorithm was tested and validated using several phantom and in vivo experiments. The enhanced images constructed by the Log-MSR algorithm were qualitatively and quantitatively analyzed in terms of brain structure and function. Our results show that the Log-MSR algorithm may provide a significant enhancement to photoacoustic imaging of deep brain tissue.

**Keywords:** photoacoustic imaging; brain; logarithmic enhancement algorithm; multi-scale Retinex algorithm; image enhancement

## 1. Introduction

Photoacoustic imaging (PAI) is an innovative biomedical imaging technique that leverages the photoacoustic effect to transform laser energy into acoustic energy through light absorption and subsequent thermal expansion [1–3]. PAI offers the dual benefits of high optical contrast and superior ultrasound penetration resolution [1,4,5]. Additionally, it allows for the quantitative determination of hemodynamic parameters such as hemoglobin, oxygen, and water concentration from multi-wavelength photoacoustic data. These parameters are crucial for providing functional information in disease diagnosis [6,7].

It is generally believed that, for most PA imaging systems, image quality remains relatively good at imaging depths of around 2–3 cm. Beyond this depth, image quality may begin to deteriorate due to increased signal attenuation, scattering, and absorption [8–10]. The photoacoustic (PA) signal quickly diminishes due to laser interaction with biological tissues, resulting in weak signals from deeper tissues and, consequently, reduced resolution and contrast in PAI [11–13]. To address this, various algorithms have been developed to enhance PAI. Liu et al. [14] introduced the photoacoustic imaging vasculature enhancement filter (PAIVEF) algorithm to boost micro-vessel imaging in rat eyes, enhancing signals while effectively suppressing noise and extending the vascular imaging depth range. Matan Benyamin et al. [15,16] employed a sparse autoencoder enhancement algorithm combined with contrast agents to improve PAI resolution. Gao et al. [17] utilized an empirical mode decomposition (EMD) algorithm with a conditional mutual information

de-noising algorithm to enhance imaging of mouse ear vascular. This algorithm improved the signal-to-noise ratio (SNR) without compromising signal fidelity or imaging speed. Lv et al. [18] merged a superpixel method with a digital mirror device (DMD) to modulate the phase and amplitude of incident light, enhancing SNR and minimizing clutter wave interference in PAI. Manwar, R. et al. [19] introduced a deep learning-based method employing a U-net neural network algorithm to match laser energy from 20 mJ/cm$^2$ to 100 mJ/cm$^2$, significantly enhancing PA image SNR and clarifying details in the lateral and third ventricles. Additionally, Manwar, R. et al. [20] developed an adaptive filter denoising method, effectively improving PA image SNR without relying on input or prior knowledge of signal characteristics, thereby reconstructing artifact-free photoacoustic images.

The enhancement algorithms previously discussed primarily focused on improving the SNR or reducing noise in overall PA signals. However, they do not address signal attenuation at increased depths. To tackle this, we employ the logarithmically local enhancement-multi-scale Retinex (Log-MSR) algorithm. The logarithmically local enhancement (Log) algorithm is designed to boost PA signals in deeper tissues by applying varying weights according to depth. Furthermore, the multi-scale Retinex (MSR) algorithm is used to refine PA images, enhancing their clarity and contrast. The efficacy of the Log-MSR algorithm was tested through several phantom and in vivo experiments, with results analyzed both qualitatively and quantitatively.

## 2. Methods

PAI encounters resolution and contrast issues at greater tissue depths due to laser attenuation, which affects signal consistency for the same object at varying depths. This hinders accurate quantitative reconstruction of PA images and highlights the need for depth-specific enhancement in PAI. In this study, we utilize the following formula to compensate for PA signals at different depths:

$$y(m, n) = A(m, n) \times x(m, n) \qquad (1)$$

where $m$ represents the horizontal distance, $n$ is the depth, $x(m, n)$ is the original image value, $A(m, n)$ is the weighting coefficient, and $y(m, n)$ is the enhanced image value [21,22]. The selection of weighting coefficients, crucial for image processing, is typically linked to PAI characteristics [21]. Finding a suitable weighting function is challenging, as mismatches can result in inadequate PA signal compensation [23]. Various functions, including step, linear, quadratic, exponential, exponential fractional power, and logarithmic functions, were tested to optimize photoacoustic image enhancement. The logarithmic function proved most effective and was therefore selected as the weighting function:

$$A(m, n) = a \times \log(n) + b \qquad (2)$$

where "$a$" adaptively adjusts the size of the logarithmic function based on the attenuation curve. "$b$" is introduced to suppress the offset in the fitted function.

An attenuation curve for the PA signal in brain tissue is depicted in Figure 1a. This curve forms the foundation for simulating PA signal attenuation in brain tissues, as illustrated in Figure 1b. The corresponding weighting function, based on the findings in Figure 1b, is shown in Figure 1c. These weightings are then applied to the original data to produce the enhanced image.

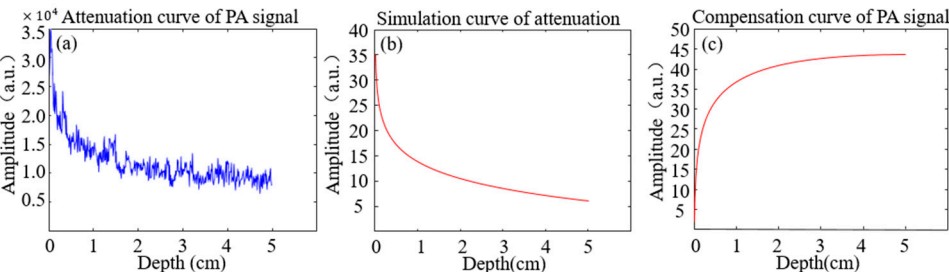

**Figure 1.** Attenuation and compensation curves for PA signal. (**a**) The attenuation with depth; (**b**) The simulated curve for attenuation with distance; (**c**) The compensating curve for weighting with distance.

To enhance the contrast of PAI, the MSR algorithm is utilized for image processing. Based on color constancy theory, MSR is an advanced image enhancement algorithm [24]. Color constancy theory suggests that colors are perceived through light-matter interactions rather than being inherent properties of objects, with object colors determined by their reflection of light across various wavelengths [25,26]. An object's color remains consistent despite variations in light [25,27]. Unlike traditional methods, Retinex maintains a balance between dynamic range compression, edge enhancement, and color constancy, allowing for versatile image enhancement [28]. By separating an image into reflection and illumination components, MSR adjusts illumination for comprehensive enhancement [29]. The Retinex theory is defined as follows [24,30]:

$$I(x,y) = L(x,y) \times R(x,y) \tag{3}$$

where $I(x, y)$ is the value of the original image, $L(x, y)$ is the illumination of the background, and $R(x, y)$ is the reflection of the object [30,31]. The MSR algorithm will help to improve the image quality by effectively removing the illumination component [32].

The MSR algorithm can be described as [30]:

$$R(x,y) = \sum_{k=1}^{n} W_k \{ \log[I(x,y)] - \log[I(x,y)^*F_k(x,y)] \} \tag{4}$$

where $n$ is the total number of Gaussian functions, $W_k$ is the weight value for each image, * is the convolution operator, and $F_k(x, y)$ is the center/wrapping function [31]. The $W_k$ satisfies the following conditions [29,33]:

$$\sum_{k=1}^{n} W_k = 1 \tag{5}$$

where $n$ is selected as 3 and evenly distribute the weights among each image, i.e., $W_1 = W_2 = W_3 = 1/3$. To achieve the center/wrapping function, we employ the Gaussian filtering function [29,30]:

$$F(x,y) = Ke^{-(x^2+y^2)/\sigma^2} \tag{6}$$

where $\sigma$ is the scale parameter. Increasing scale results in a richer coordination of colors within the image [32,33]. Conversely, decreasing scale enhances the clarity of textures in the image [32,33]. In this study, $\sigma$ is set at 15, 80, and 200, respectively. Meanwhile, the normalization constant K is determined to meet the following Equation (7) [31]:

$$\iint F(x,y)dxdy = 1 \tag{7}$$

MSR, by using multiple scales, is a robust image processing technique. Compared to the single-scale Retinex algorithm, MSR not only preserves image textures but also maintains imaging fidelity [29,32].

## 3. Materials

### 3.1. Phantoms and Animals

In this study, we conducted both phantom and in vivo experiments to assess the Log-MSR algorithm. For the phantom experiment, objects consisted of a mixture of 2 g agar powder, 98 mL water, and 5 μL India ink. This mixture was injected into cylindrical molds measuring 3 mm in diameter and 3 cm in height and then cooled. The background model was created using 8 g of agar powder, 8 mL of lipid emulsion (C6–C24), and 384 mL of water, formed into a rectangular prism of 10 cm length, 7 cm width, and 3 cm height [34]. The objects were embedded into the background phantom at varying depths of 1 cm, 3 cm, and 5 cm from the boundary.

In the animal experiments, SPF-grade Sprague Dawley (SD) rats, averaging 7 weeks old and weighing approximately 220 g, were sourced from Dashuo Biotechnology Corporation (Chongqing, China).

Prior to hemorrhage experiments, the rats' heads were depilated. The rats were anesthetized with 2% sodium pentobarbital via intraperitoneal injection at a dosage of 2 mL/kg and positioned prone on a rat board, heads aligned forward. The head was secured in a stereotaxic instrument, adjusted to align the bregma and lambda horizontally. After disinfection, a 2 cm incision along the skull's midline exposed the skull by peeling off the periosteum, with the bregma serving as a reference point. A mark was placed 2.0 mm to the right and 1.5 mm posterior to the bregma, where a hole was drilled until the dura mater was visible. A microinjection syringe, filled with type VII collagenase and attached to the stereotaxic instrument, was carefully inserted vertically into the hole to a depth of 5 mm, near the caudate nucleus. Collagenase (0.5 μL) was injected at a rate of 200 nL/min over 5 min. After the injection, the needle remained in place for 10 min before slowly withdrawing. The drilled hole was sealed with sterile bone wax to prevent infection, followed by disinfection and suturing of the incision.

### 3.2. Photoacoustic Imaging System

In this study, we employed a real-time, fast PAI system for both phantom and animal experiments. Figure 2 presents a schematic diagram of the system. The PAI system's excitation source is a high-frequency pulsed laser (SpitLight EVO S OPO, Innolas, Justus-von-Liebig Ring 8, 82152 Krailling, Germany, pulse width: 5–10 ns, repetition frequency: 100 Hz, adjustable wavelength range: 680–980 nm). The laser beam, channeled through an optical fiber with a 1 mm × 60 mm light outlet and an energy density of approximately 2 mJ/cm$^2$, irradiates the target object. A semi-circular transducer with 256 arrays (ULSO Electronics Technology Co., Ltd., Xingtai city, Hebei, China; detector curvature radius: 80 mm; center frequency: 3.2 MHz) is mounted on a gimbaled robotic arm positioned on a three-dimensional translation stage. The fiber's outlet is positioned above the transducer, which is encased in water-filled translucent plastic sheeting (TPU) with transabdominal ultrasound properties. PA signals are amplified and filtered by the data acquisition (DAQ) system (sampling rate: 40 MSPS, input bandwidth: 50 KHz~20 MHz, adjustable gain: 54~80 dB, sampling points: 512~4096 p/frame), triggered by laser pulse synchronization. Wavelengths of 760 nm, 840 nm, and 930 nm were used in vivo, with a wavelength switching time under 10 ms and incident energy density maintained below 10 mJ/cm$^2$. The delay and sum (DAS) algorithm reconstructed the PA images, achieving a spatial resolution of about 100 μm and a temporal resolution of about 0.01 s. All calculations for evaluating the Log-MSR algorithm's performance and accuracy were performed on a personal computer (PC, DELL G15 5520, Round Rock, TX, USA) equipped with an Intel I7-12700H CPU, 2.7 GHz, and 16 GB of RAM.

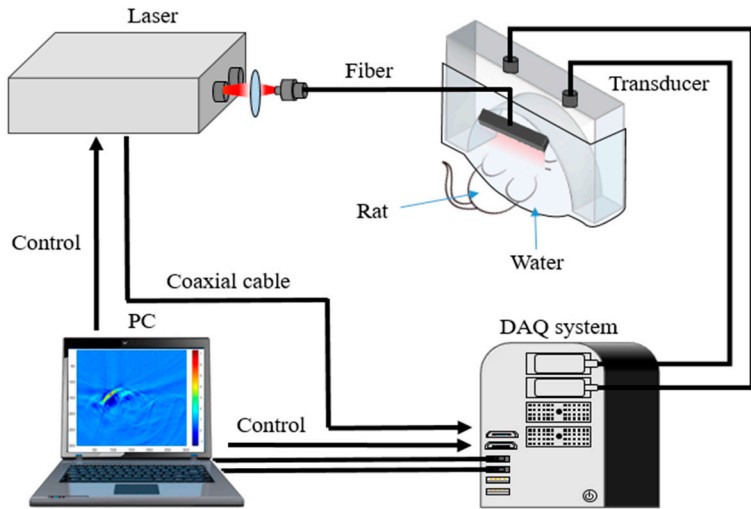

**Figure 2.** Real-time, fast PAI system.

## 4. Results and Discussion

### 4.1. Phantom Experiments

The phantom experiments were conducted to validate the Log-MSR algorithm, as depicted in Figure 3. Figure 3a–e displays, respectively, the cross-section photograph of the phantom, the original PA image, the logarithmic algorithm-enhanced PA image, the MSR algorithm-enhanced PA image, and the Log-MSR algorithm-enhanced PA image.

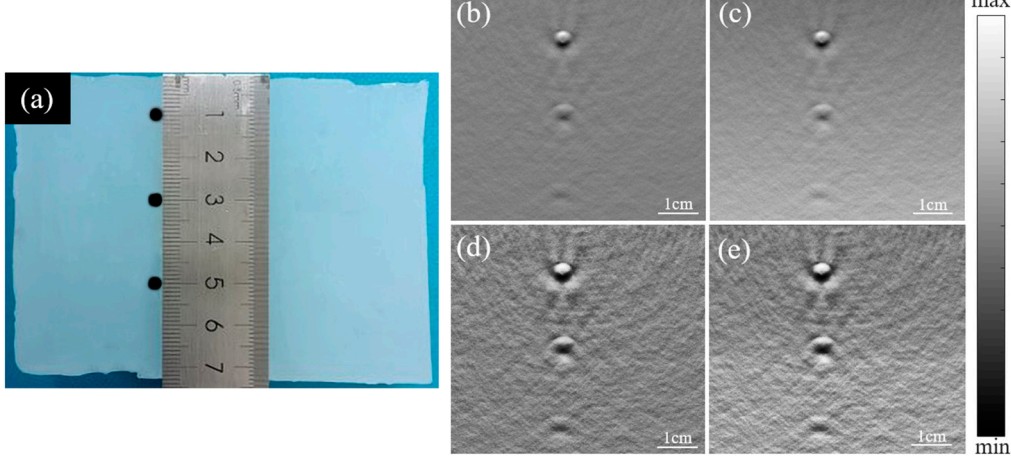

**Figure 3.** Photograph and reconstructed PA images of three objects embedded in a background phantom. (**a**) Cross-section photograph of the phantom; (**b**) original PA image; (**c**) logarithmic algorithm-enhanced PA image; (**d**) MSR algorithm-enhanced PA image; (**e**) Log-MSR algorithm-enhanced PA image.

In Figure 3b, the original image displayed reduced contrast at increased depth. However, in Figure 3c, the logarithmic algorithm-enhanced image exhibited a marked increase in contrast at depth. Additionally, Figure 3d shows the MSR algorithm-enhanced image with clearer boundaries. Figure 3e presents the image enhanced by the Log-MSR algorithm, which demonstrates a more substantial enhancement compared to images enhanced by a single algorithm. The Log-MSR algorithm effectively compensated for the PA signals and significantly improved imaging contrast and sharpness.

The diameters of three objects, positioned from top to bottom in Figure 3b,e, were measured and are presented in Figure 4. Figure 4b and 4d show enlarged views of the three

objects from Figure 4a and 4c, respectively. Figure 4e,f depicts the signal profiles of the images in Figure 4a,c, marked by red lines.

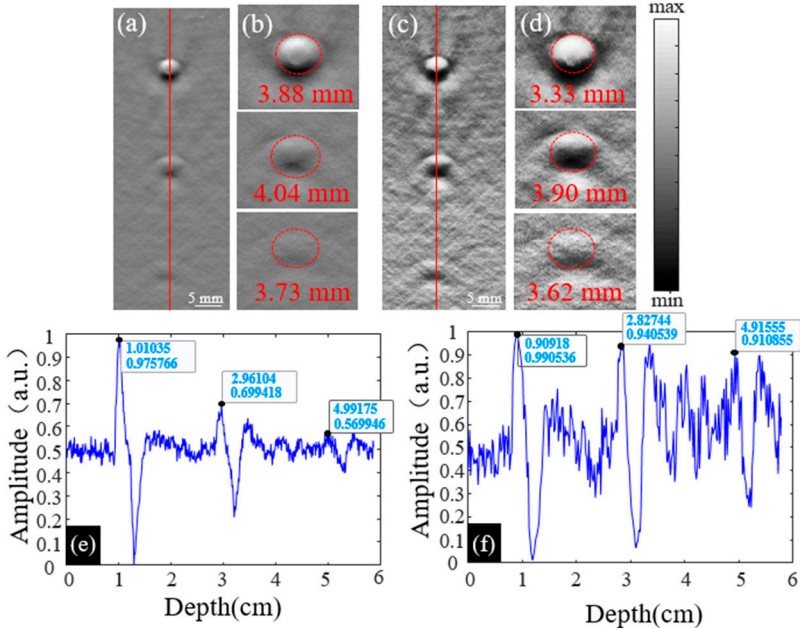

**Figure 4.** (**a**) The original PA image of three objects; (**b**) the enlarged PA images of objects at 1 cm, 3 cm, and 5 cm in (**a**); (**c**) the enhanced PA image of objects constructed by the Log-MSR algorithm; (**d**) the enlarged PA images of objects at 1 cm, 3 cm, and 5 cm in (**c**); (**e**) the line profile of the original PA image in (**a**), the profile is pointed out by a red line; (**f**) the line profile of the original PA image in (**c**), the profile is pointed out by a red line.

The diameters of each object in Figure 4b,d are shown in Table 1. At depths of 1.0 cm, 3.0 cm, and 5.0 cm, the diameter errors between the actual object and the original image were 29.33%, 34.67%, and 24.33%, respectively. At the same depths, the errors between the actual object and the Log-MSR algorithm-enhanced image were 11.00%, 30.00%, and 20.67%, respectively.

**Table 1.** The measured diameter of objects at various depths before and after enhancement.

| Object's Location | 1 cm | 3 cm | 5 cm |
|---|---|---|---|
| Original algorithm | 3.88 | 4.04 | 3.73 |
| Error 1 | 29.33% | 34.67% | 24.33% |
| Log algorithm | 3.88 | 4.04 | 3.67 |
| Error 2 | 29.33% | 34.67% | 22.33% |
| MSR algorithm | 3.46 | 3.95 | 3.56 |
| Error 3 | 15.33% | 31.67% | 18.67% |
| Log-MSR algorithm | 3.33 | 3.90 | 3.62 |
| Error 2 | 11.00% | 30.00% | 20.67% |

Actual diameter of object: 3 mm.

In Figure 4a,c, line profiles were derived by extracting the PA signal along the red line, with results shown in Figure 4e,f. Table 2 displays the PA signal values for each object in these figures. Compared to the object at 1.0 cm depth in the original image, the values decreased by 14.29% and 15.81% at 3.0 cm and 5.0 cm depths, respectively. In contrast, for the Log-MSR algorithm-enhanced image, the values decreased only by 3.80% and 8.01% at the same depths. The signal values at different depths in the Log-MSR enhanced image were nearly identical, demonstrating the algorithm's effective compensation of PA signals in deep tissue.

**Table 2.** The signal intensity of the objects at depths of 1 cm, 3 cm, and 5 cm.

| | Original Algorithm | | | Log-MSR Algorithm | | |
|---|---|---|---|---|---|---|
| Object's location | 1 cm | 3 cm | 5 cm | 1 cm | 3 cm | 5 cm |
| Average amplitude | 0.6156 | 0.5276 | 0.55183 | 0.6916 | 0.6653 | 0.6362 |
| Attenuation ratio | - | 14.29% | 15.81% | - | 3.80% | 8.01% |

Unit of signal amplitude: a.u.

*4.2. Animals*

The results from animal experiments are depicted in Figure 5.

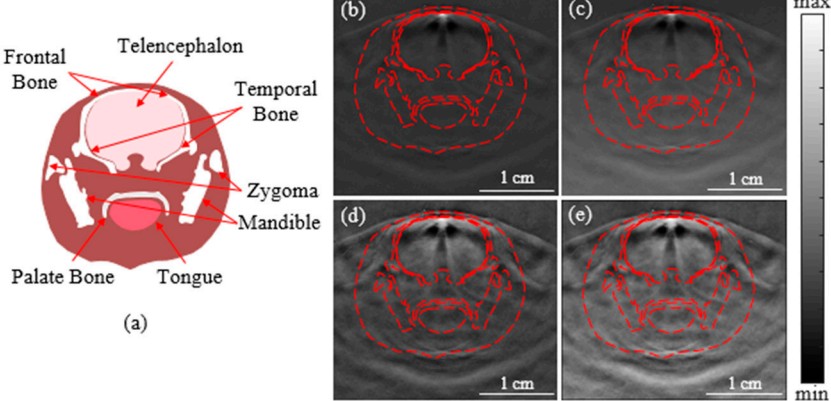

**Figure 5.** Diagram and PA images of the rat brain. (**a**) Diagram of cross-sectional rat brain; (**b**) the original PA image; (**c**) the enhanced image constructed by the logarithmic algorithm; (**d**) the enhanced image constructed by the MSR algorithm; (**e**) the enhanced image constructed by the Log-MSR algorithm.

In Figure 5b, the contrast of the original images diminished with increasing depth. The image enhanced by the Log-MSR algorithm (Figure 5e) exhibited significantly stronger enhancement in both contrast and depth compared to those enhanced by a single algorithm (Figure 5c,d). The Log-MSR algorithm not only effectively compensated for the PA signals but also dramatically improved imaging contrast and clarity.

The contrast-to-noise ratio (CNR) of the four images was calculated to quantify the enhanced imaging quality. The results are shown in Table 3. CNR indicates the reconstruction effect of the ROI in the image. The calculation formula is as follows [35,36]:

$$CNR = \frac{\mu_{ROI} - \mu_{Back}}{\left( \sigma_{ROI}^2 \frac{A_{ROI}}{A_{Total}} - \sigma_{Back}^2 \frac{A_{Back}}{A_{Total}} \right)^{1/2}} \qquad (8)$$

where $\mu_{ROI}$ is the expectation of the object, $\mu_{Back}$ is the expectation for the background, $\sigma_{ROI}^2$ is the variance of the object, $\sigma_{Back}^2$ is the variance of the background, $A_{ROI}$ is the area of the object, $A_{back}$ is the area of the background, and $A_{Total}$ is the area of the total image [35,36].

**Table 3.** PA image contrast of rat brain.

| | Original Algorithm | Log Algorithm | MSR Algorithm | Log-MSR Algorithm |
|---|---|---|---|---|
| CNR | 0.1649 | 0.1337 | 0.4882 | 0.3981 |

Unit of contrast: a.u.

Table 3 reveals that the MSR algorithm significantly improved image contrast. However, it failed to compensate for PA signals in deep brain tissue. The contrast in deep brain tissue was markedly lower compared to shallow brain tissue in the MSR-enhanced image (Figure 5d). Compared to the original image, the Log algorithm reduces the contrast

of the image by 0.81 times, while the MSR algorithm increases the contrast of the image by 2.96 times. The Log-MSR algorithm, on the other hand, enhances the contrast of the image by 2.41 times. Additionally, the PA signals in deep brain tissues (Figure 5e) were enhanced following logarithmic treatment. The Log-MSR algorithm combined the benefits of logarithmic local compensation and MSR algorithms, enhancing image contrast and compensating for PA signal attenuation with depth.

To evaluate the Log-MSR algorithm's performance in pathological imaging, it was applied to cerebral hemorrhage experiments. The algorithm's effectiveness was assessed by examining changes in morphological structures and hemodynamics within the hemorrhage area before and after enhancement.

In the hemorrhage experiments with two groups of rats, the results are shown in Figure 6. The original images (a,d) displayed poor contrast and unclear boundaries in the hemorrhagic areas. The Log-MSR algorithm significantly improved the contrast of the enhanced images, with the hemorrhagic areas distinct from the surrounding normal brain tissue. The areas of brain hemorrhage were quantified in the PA images before and after enhancement and compared with the corresponding areas in actual brain sections, as detailed in Table 4.

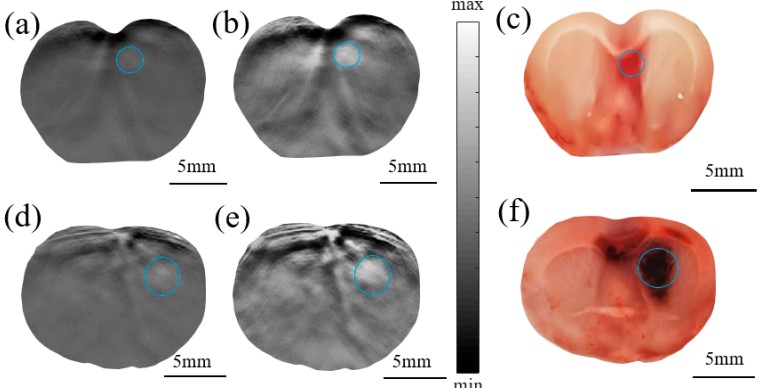

**Figure 6.** PA original image (**a**) and the enhanced image constructed by the Log-MSR algorithm (**b**) and photograph of the actual brain section (**c**) of rat #1; PA original image (**d**) and the enhanced imaging constructed by the Log-MSR algorithm (**e**) and photograph of the actual brain section (**f**) of rat #2. (The hematoma areas are shown in the blue circles).

**Table 4.** Area of cerebral hemorrhage.

| | NO.1 Rat | Error 1 | NO.2 Rat | Error 2 |
|---|---|---|---|---|
| Photograph of the actual brain section | $3.0128 \pm 0.1962$ | - | $7.2716 \pm 0.3154$ | - |
| Original image | $2.5536 \pm 0.2482$ | 15.24% | $6.6079 \pm 0.3946$ | 9.13% |
| Enhanced image constructed by the Log-MSR | $2.7692 \pm 0.1062$ | 8.09% | $7.0923 \pm 0.3456$ | 2.47% |

Units of area: $mm^2$.

The hemorrhagic area was approximated as an ellipse, measured by the major axis '*a*' and minor axis '*b*'. Values of '*a*' and '*b*' were determined using the half-width-at-full-maximum (HWFM) of the PA signal along the length and width of the hemorrhage area, relative to the global average. This method parallels the common approach for calculating hematoma area in CT imaging [37]. Hence, the hemorrhage area in our study was calculated as follows:

$$Area(ICH) = \frac{\pi}{4}ab \tag{9}$$

Table 4 summarizes the errors in estimating the cerebral hemorrhage area. The discrepancies between the original image and the actual brain slice were 15.24% and 9.13%,

respectively. In contrast, the errors between the Log-MSR algorithm-enhanced image and the actual slice were significantly lower, at 8.09% and 2.47%, respectively. These findings indicate that the Log-MSR algorithm substantially enhanced the contrast of PAI, demonstrating its potential to improve both the quality and diagnostic accuracy of PAI.

Each substance in biological tissue possesses a unique optical absorption coefficient. PAI leverages this principle to measure the optical absorption of endogenous contrast agents within brain tissue [1]. In this study, a multi-wavelength algorithm employing wavelengths of 760 nm, 840 nm, and 930 nm was used to assess brain tissue hemodynamics, including oxygen concentration ($sO_2$), oxygenated hemoglobin concentration ($HbO_2$), deoxyhemoglobin concentration (HbR), and water concentration ($H_2O$) [4,6]. As depicted in Figure 7, hemodynamic images were acquired from both the normal group (shown in Figure 7a–d) and the cerebral hemorrhage group (shown in Figure 7i–l) and subsequently enhanced to produce images for the enhanced control group (Figure 7e–h) and the cerebral hemorrhage group (Figure 7m–p).

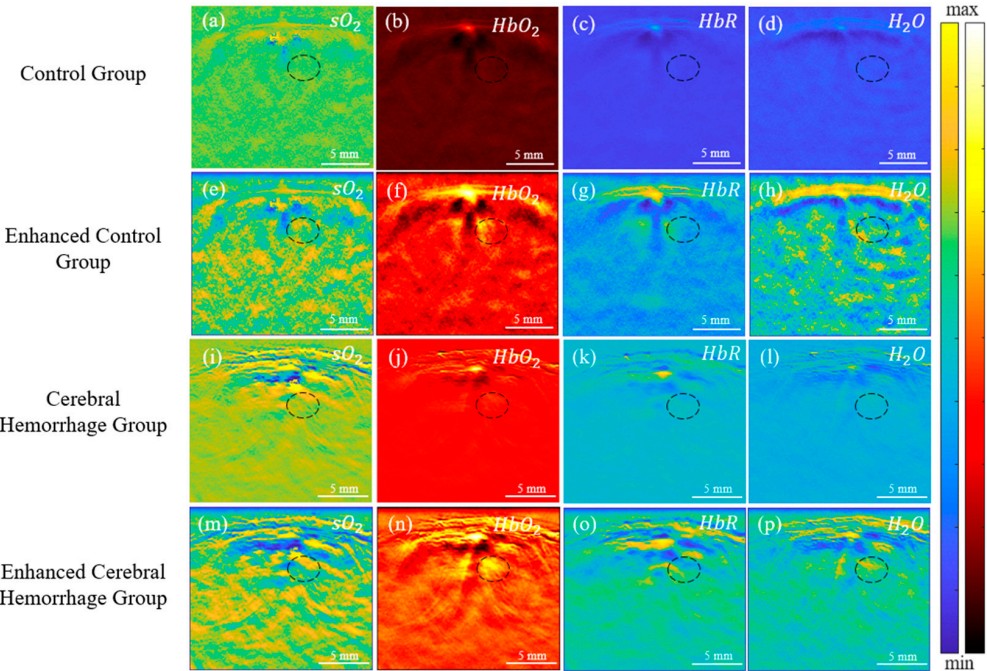

**Figure 7.** In vivo PA images of rat brain hemodynamics for both the control and cerebral hemorrhage groups. (**a–d**): PA images for the control group; (**e–h**): PA images for the enhanced control group; (**i–l**): PA images for the cerebral hemorrhage group; (**m–p**): PA images for the enhanced cerebral hemorrhage group. (The hematoma area is shown in the black circle).

The accumulation of blood and subsequent hematoma formation in the affected area led to increased optical absorption of chromophores. The results in Figure 7 reveal enhanced contrast in $HbO_2$, HbR, and $H_2O$ in the hemorrhage area. Figure 7i,m shows decreased $sO_2$ contrast within the hematoma lesions post-intracerebral hemorrhage (ICH) surgery. In the hematoma region, there was an increase in HbR and a decrease in $HbO_2$, culminating in a significant decrease in $sO_2$ [38].

The black oval in Figure 7 shows the location of the cerebral hemorrhage. From the results, a significant decrease in $sO_2$ of the hemorrhagic area in the cerebral hemorrhage group was observed, while $HbO_2$, HbR, and $H_2O$ had a significant increase. Following enhancement by the Log-MSR algorithm, significant changes in HbR were observed in the enhanced cerebral hemorrhage group compared to the normal group, and there were also slight differences in other parameters. The average value of the hemorrhagic areas was calculated for the normal group, intracerebral hemorrhage group, enhanced normal

group, and enhanced intracerebral hemorrhage group, respectively. The results are shown in Figure 8.

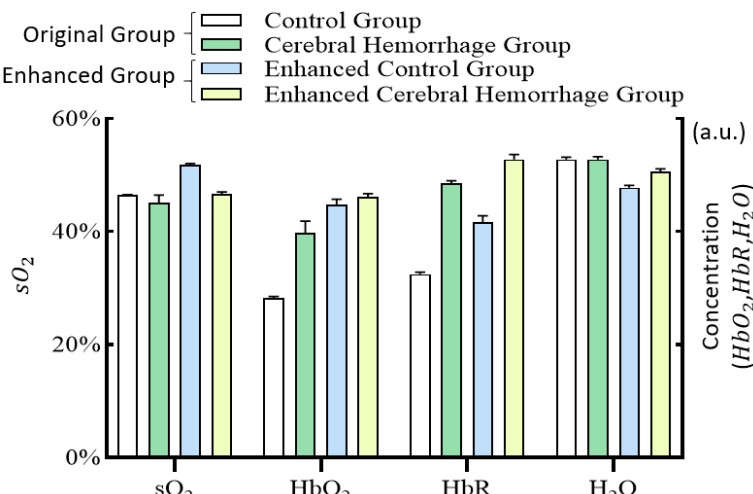

**Figure 8.** Histogram of hemodynamics in the cerebral hemorrhage area for the control group, the cerebral hemorrhage group, the enhanced control group, and the enhanced cerebral hemorrhage group.

Figure 8 illustrates that $sO_2$ levels in the cerebral hemorrhage group were lower than those in the normal group, while $HbO_2$, HbR, and $H_2O$ levels were higher. These findings align with previous pathological analyses. After enhancement by the Log-MSR algorithm, the changes in $sO_2$ and $H_2O$ in the enhanced group were more pronounced, whereas changes in $HbO_2$ and HbR were less marked compared to the original image.

## 5. Conclusions

In this study, we introduced the Log-MSR algorithm and applied it to enhance PAI performance for deep tissue imaging. Our phantom and animal experiments successfully demonstrated the algorithm's feasibility and effectiveness in enhancing image quality, specifically in terms of clarity and contrast improvement. Additionally, PAI signal attenuation remained below 10% with increasing depth. The algorithm's efficacy was further evidenced in a cerebral hemorrhage experimental study in rats, reducing the error in the hemorrhagic area measurement in the Log-MSR-enhanced image to 2.47%. Overall, the Log-MSR algorithm holds promise for advancing the application of PAI in deep tissue imaging.

**Author Contributions:** Conceptualization, Y.X. and H.J.; methodology, D.W.; software, Y.X.; formal analysis, Y.W. (Yanting Wen) and J.Z.; investigation, X.W.; resources, Y.C.; data curation, Y.X. and Y.Y.; writing—original draft preparation, Y.X.; writing—review and editing, Y.X., D.W. and H.J.; visualization, Y.X. and Y.W. (Yun Wu); supervision, Z.C.; project administration, D.W. and H.J.; funding acquisition, D.W. and H.J. All authors have read and agreed to the published version of the manuscript.

**Funding:** This work was supported by grants from Youth Fund of the National Natural Science Foundation of China, grant number 6220012213 and the Chongqing post-doctoral research project (special funding project), grant number 2021XM3040.

**Institutional Review Board Statement:** The study was conducted in accordance with the Declaration of Helsinki and approved by Chongqing University of Posts and Telecommunications (protocol code: NSFCLL202305; date of approval: 15 March 2023).

**Informed Consent Statement:** Not applicable.

**Data Availability Statement:** Data is unavailable due to privacy restrictions.

**Conflicts of Interest:** The authors declare no conflict of interest.

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
