# Peer review of "Image Enhancement Method for Photoacoustic Imaging of Deep Brain Tissue"

_photonics, doi:10.3390/photonics11010031_

Round 1

Reviewer 1 Report

Comments and Suggestions for Authors

The authors introduced the imaging process algorithm, multi-scale retinex (MSR), for photoacoustic imaging of deep tissue.  The results were obtained from in vitro and in vivo studies. Interesting approach was introduced but, the study is weak to convince of the efficiency of the application. The study is focused on the improved image in deep tissue.  It should show how improved depending on the depth with reasonable analysis and the proof.

1.       At the first word in Introduction, PAI means photoacoustic imaging. The full letters of the word are shown in abstract. But could you show the meaning of abbreviations at the first use in manuscript?

2.       Introduction, line 36, “PAI is not ideal” for imaging deep tissues.  It would be better to define how deep it means.

3.       Line 49, what is the meaning of DMD?

4.       The manuscript has wrong information related with references.

-          Reference 13, and 14 , DOI information is same.

-          Referenc 16 has problems (ex fig1).  Difficult to understand  I recommend to change it to another.

5.       Line 72, what was the multiple experiments? Were the results obtained by authors or by reference? It looks the result is Fig 1.  When the result in Figure 1a (if the first left image is a) is derived from brain tissue, how was the experiment done?

6.       The Methods part must explain more details

-          In Materials, phantom was made using agar.  The 2 g of agar with 98ml of water and ink was used. The phatom should be very soft.  Why is the recipe used? The Indian ink was mixed with water and agar? Or is it used for pointing black dots like fig3a?  If it is, the phantom light attenuation is same with real tissue?

-          And, the 7 week-old rat was used. What strain is it? And what does it mean, “the rat’s head was made a skin preparation”?

-          There is no method explanation about hemorrhage animal.

7.       In phantom result, the signal from deep target was increased but SNR looks nothing improved. The noisy was also amplified. Is any proof of better SNR in deep area?

Comments on the Quality of English Language

Some expression is not understandable. 

Reviewer 2 Report

Comments and Suggestions for Authors

The photoacoustic signals from brain tissue are significantly weakened by skull distortion effects, leading to reduced resolution and contrast. To overcome this challenge, a Logarithmic Multi-Scale Retinex (Log-MSR) algorithm is introduced in this paper. This algorithm combines the Logarithmic Depth Enhancement (Log) algorithm, which compensates for signal attenuation at different depths, with the Multi-Scale Retinex (MSR) algorithm, designed to enhance image contrast.

Despite offering new insights, major revisions are required before this paper can be considered for publication. Below, I provide my comments on this paper:

• Introduction: It is recommended to incorporate more recent state-of-the-art image reconstruction algorithms for deep tissue imaging in the introduction, as they are relevant to the context of enhancing PA signals for deep tissue imaging. Please add relevant literature, see: Deep learning protocol for improved photoacoustic brain imaging, by Rayyan Manwar et al. and Photoacoustic signal enhancement: towards utilization of low energy laser diodes in real-time photoacoustic imaging, by Rayyan Manwar et al.

• Weighting Function: Clarify the weighting function used before adopting the logarithmic weighting function, as mentioned after multiple experiments.

• Figure 1.a: Provide additional information on how Figure 1.a is extracted and the methodology behind its production.

• Line 81: Elaborate on the sensitivity of the algorithm to the basis curve used for simulating the attenuation of PA signals in brain tissues. Consider discussing the implications of using alternative bases.

• Color Constancy Theory: Since the developed algorithm is based on the color constancy theory, consider adding more summary information about this theory for a better understanding.

• Line 143 and 144: Correct the typo, it should be DAS, not DAM.

• Color Scale: Ensure the use of a proper text for the color scale in all figures. Go through each figure and make necessary modifications.

• Figure 4.b and 4.d: Clarify the criteria for selecting the border of the red circle in these figures.

• Table 1: Include information about the approaches used to generate Figure 3.c and 3.d. I cannot see any specific difference between 3.d and 3.e visually.

• Table 2: For a more accurate comparison, suggest comparing the average intensity within each target rather than just the first 3 picks of the signal.

• Fig 5.a: Remove the red underline in Figure 5.a.

• Equation (8): Specify the reason for choosing Contrast Enhancement ratio over Contrast-to-Noise Ratio (CNR). Clarify whether CNR or Contrast Enhancement ratio (CER) was calculated.

• I find the numbers in Table 3, such as 196.06% and 141.42%, strange. To facilitate a more meaningful comparison, consider employing alternative quantitative metrics.

Comments on the Quality of English Language

Language is good. 

Reviewer 3 Report

Comments and Suggestions for Authors Minor comments:  

1. The definition of the hemorrhage area seems random to me. Why pixels with values above the global mean are considered hemorrhage area?

2. Why were 760 nm840 nm and 930 nm chosen for illumination?

3. How are the errors (Table 1), attenuation ratio (Table 2) and enhancement ratio (Table 3) calculated in the paper?

4. Please give details of the values of the molar extinction coefficients used for the calulations.

Comments on the Quality of English Language

 Minor editing of English language required

Round 2

Reviewer 1 Report

Comments and Suggestions for Authors

The revised manuscript has much better quality. I would like to recommend the paper published in Photonics.